# Exploring Hypernetwork to Enhance Model Heterogeneous Personalized Federated Learning with Data Distillation

## Abstract

Personalized federated learning (pFL) aims to provide each client with a customized model based on global knowledge. However, in highly heterogeneous scenarios, pFL often struggles to obtain effective global information and faces a trade-off between personalization and generalization, which can degrade overall generalization performance. To address this issue, we propose a **M**odel-**H**eterogeneous **p**ersonalized **Fed**erated learning framework based on **H**yper**N**etworks with **D**ata **D**istillation, **MH-pFedHNDD**, which, for the first time, incorporates data distillation into a hypernetwork-based federated learning framework, introducing a data-driven perspective to tackle this problem. We design two effective regularization terms: (1) *Contrastive Condensation Loss*, which encourages the latent embeddings of synthetic data to be more compact and closely aligned with the local data of clients used as anchors; (2) *Reg Loss*, which integrates the latent embeddings of all clients' synthetic data as anchors to guide the optimization direction for generalization, thereby enhancing each client's personalized optimization performance on its local data along with the use of universum negatives. By leveraging synthetic data distilled with more robust global information, our method enhances local training on clients, is the first to alleviate the imbalance between commonality and personalization for hypernetworks, and improves the performance and generalization of the hypernetwork. Extensive experiments under various settings demonstrate the effectiveness of our MH-pFedHNDD in personalized federated learning. Our code is available at `https://anonymous.4open.science/r/MH-pFedHNDD`.

## 1 Introduction

Federated learning (FL) (McMahan et al., 2017; Yang et al., 2019) is a general distributed machine learning paradigm that has been widely applied by researchers and engineers across various domains (Zhou et al., 2024; Murmu et al., 2024; Feng et al., 2024). However, a single global model often fails to meet the needs of all clients due to non-IID data. To address this, personalized federated learning (pFL) (Smith et al., 2017; T Dinh et al., 2020; Deng et al., 2020) has been proposed, which allows each client to maintain a personalized model rather than a shared global model, thereby better accommodating client-specific tasks and data distributions (McMahan et al., 2016).

In practice, current pFL faces the practical settings of model heterogeneity (Chen et al., 2023a), as the devices running local clients typically differ in computational resources (Chai et al., 2020; Shin et al., 2024), communication capabilities (Caldas et al., 2018; He et al., 2020; Shah & Lau, 2021), and model architectures (Li & Wang, 2019; Zhu et al., 2021; Wu et al., 2024). Meanwhile, to prevent privacy leakage, model-heterogeneous pFL also needs to address the challenges of preserving both data (Shokri et al., 2017) and model (Zhang et al., 2024a) privacy.

For this issue, model-heterogeneous pFL (Kulkarni et al., 2020; Tan et al., 2022) has primarily focused on incorporating methods such as data distillation, hypernetworks, and model decoupling to tackle the problem of model heterogeneity. Model decoupling (Jang et al., 2023; Yi et al., 2023a;b) splits the local model on clients into a feature extractor component and a classifier component. However, this low-level knowledge sharing can hinder client collaboration and negatively impact overall

performance. Although data distillation approaches (Huang et al., 2024) can integrate data from multiple clients, they overlook model structure considerations, resulting in insufficient generalization for model heterogeneity. Hypernetwork-based FL approaches (Shamsian et al., 2021; Zhu et al., 2023; Scott et al., 2024) utilize a hypernetwork on the server to generate personalized models for clients. Nevertheless, these methods overlook the statistical information of data distributions across different clients, resulting in suboptimal accuracy.

In fact, the coexistence of model heterogeneity and data heterogeneity (T Dinh et al., 2020; Lin et al., 2020; Li et al., 2022; Liu et al., 2024) poses significant challenges to model-heterogeneous pFL. Therefore, an ideal approach could leverage hypernetworks to address model heterogeneity, while also striking a balance between the individuality and commonality of data distributed across different clients (Tang et al., 2021; Chen et al., 2023c; Mclaughlin & Su, 2024; Cui et al., 2024; Pan et al., 2025), ultimately achieving well-performing models. However, this requires clients to share either data or models to convey their statistical information, which may compromise privacy.

To address the challenges mentioned above, we are the first to combine hypernetworks with data distillation to tackle the problem of model-heterogeneous pFL, without requiring the sharing of raw local data or models. We propose **M**odel-**H**eterogeneous **p**ersonalized **Fed**erated learning framework based on **HyperN**etworks with **D**ata **D**istillation (**MH-pFedHNDD**), which adopts a data-driven perspective to address this challenge. In our approach, each local client distills its local data once and uploads it to the server. The server then aggregates synthetic data from all clients to form a synthetic dataset, which assists local client training. This process enables the sharing of global commonalities while maintaining local individuality, achieving a better balance between personalization and generalization. In addition, we carefully design two effective regularization terms: (1) Contrastive Condensation Loss during synthetic data generation, which encourages the latent embeddings of synthetic data to be more compact and closely aligned with the local data of clients used as anchors, generating high-quality synthetic data; (2) Reg Loss during client training with the synthetic dataset, which integrates the latent embeddings of all clients' synthetic data as anchors to guide the optimization direction of each client for generalization, then achieves better local training results and ultimately enables the hypernetwork to generate higher-quality personalized models with the help of universum negatives (UniNegs). At the same time, we follow the state-of-the-art approach (Zhang et al., 2025) by enabling hypernetworks to generate parameters for clients with similar parameter sizes with the same designed heads, thereby boosting performance and reducing computational overhead. Our framework ultimately addresses both data and model heterogeneity by combining hypernetworks with a novel data-driven perspective, and is the first to achieve improved performance and a better balance between commonality and individuality for the hypernetwork, all while preserving privacy. Our major contributions are summarized as follows:

- We are the first to propose a data-driven perspective that enables a hypernetwork-based FL solution for model heterogeneity in personalized federated learning. Our method, MH-pFedHNDD, is the first to integrate data distillation into the hypernetwork framework as well as better balance personalization and generalization under data heterogeneity.

- We introduce Contrastive Condensation Loss, which serves as an anchor to better align synthetic data with local data during the distillation, and Reg Loss, which enhances the generalization of the local model and further boosts its personalized performance during the training by combining with the universum negatives.

- We conduct extensive experiments on three widely used datasets with various state-of-the-art baselines. The results demonstrate that our method consistently outperforms the baselines across different tasks, validating the effectiveness of our approach.

## 2 RELATED WORK

### 2.1 FEDERATED DATA DISTILLATION

Dataset Distillation (Wang et al., 2018) can be applied to data by matching outputs or gradients (Zhao et al., 2020a; Zhao & Bilen, 2021; Wang et al., 2022; Cazenavette et al., 2022), which helps generate a small synthetic dataset from a large dataset to reduce the impact of data heterogeneity in FL (Wang et al., 2024a; Jia et al., 2024).

Goetz & Tewari (2020) are the first to introduce data distillation methods into FL, where, instead of transmitting gradients back to the server, a small amount of synthetic data is communicated, thereby reducing communication costs DOSFL (Zhou et al., 2020) and FedD3 (Song et al., 2022) explore data distillation to improve performance in the complex one-shot Federated Learning settings (Liu et al., 2025). FedDM (Xiong et al., 2023) implements a DM-based data condensation (Zhao & Bilen, 2023) on the client side, with the server using condensed data from clients to approximate the original global training loss in FL; FedAF (Wang et al., 2024b) reduces client drift and boosts model performance via peer knowledge and condensed data. However, existing methods cannot be deployed in personalized FL settings. Our MH-pFedHNDD integrates the effective distribution matching method DM (Zhao & Bilen, 2023) to, for the first time, enhance the performance of hypernetworks from a data-driven perspective.

## 2.2 Personalized Federated Learning

To account for the model heterogeneity (Chen et al., 2023a) and data heterogeneity (T Dinh et al., 2020; Lin et al., 2020; Li et al., 2022; Liu et al., 2024) in pFL, HeteroFL (Diao et al., 2021) divides the global model into different sub-models for clients based on computational capabilities, thus allowing the local models. FedRolex (Alam et al., 2022) utilizes a rolling scheme that allows for training different sub-models for the global model evenly. FedClassAvg (Jang et al., 2023) shows its effectiveness by employing classifier weights as an agreement to help clients learn about scarce labels FedGH (Yi et al., 2023a) uses a generalized global prediction header for diverse model structures, while FedTGP (Zhang et al., 2024b) leverages prototype learning with their Adaptive-margin-enhanced Contrastive Learning to learn the trainable global prototype features on the server to improve the accuracy.

Hypernetworks (Ha et al., 2017) have recently been recognized as a promising solution for pFL, where a single network is employed to generate personalized model parameters from clients' embedding vectors of local models (Shamsian et al., 2021; Zhu et al., 2023; Scott et al., 2024) and output the weight ratio during aggregation (Ma et al., 2022). pFedHN (Shamsian et al., 2021) introduces a hypernetwork to directly produce a personalized model. pFedLHN (Zhu et al., 2023) leverages a layer-wise hypernetwork to achieve fine-grained personalization across different layers of the model. MH-pFedHN (Zhang et al., 2025) quantifies clients with different architectures and generates parameters using customized embedding vectors. In their design, clients with identical parameters share the same customized embedding vector and the same designed heads. MH-pFedHNGD (Zhang et al., 2025) extends MH-pFedHN by introducing a lightweight plug-in global model to improve the final results from a model-driven perspective. In contrast, our MH-pFedHNDD tackles the pFL problem from a data-driven perspective and achieves a better balance between commonality and personalization, which the above approaches fail to address.

## 2.3 Commonality and Personalization in Personalized Federated Learning

In pFL, achieving commonality of global model while preserving personalization of local models is a non-trivial problem. For example, approaches such as local fine-tuning, FedPer (Arivazhagan et al., 2019a), Ditto (Li et al., 2020), Per-FedAvg (Fallah et al., 2020b) and FedRep (Collins et al., 2021) all emphasize retaining local characteristics. However, excessive personalization can undermine commonality, thereby degrading the generalization ability of the model for data from new labels.

Some methods (Fallah et al., 2020a; Mansour et al., 2020; Tan et al., 2022) first train a shared base model on the server and then perform additional training on local data to achieve personalization. Other methods (Deng et al., 2020; Qi et al., 2025) introduce regularization terms related to the global model during local training to strike a balance between commonality and personalization. However, in highly heterogeneous environments, effective global information may be unavailable; furthermore, the balance between personalization and commonality may be disrupted, leading to excessive personalization and reduced generalization ability of the model (Arivazhagan et al., 2019b; Chen et al., 2023b; Tran et al., 2025). DESA (Huang et al., 2024) enhances the generalization of decentralized federated learning by using synthetic anchors for effective knowledge transfer. However, none of the previous methods can be directly applied to hypernetworks. Our MH-pFedHNDD is the first to achieve both commonality and personalization within a hypernetwork framework, while further enhancing performance through the use of universum negatives (Han et al., 2022).

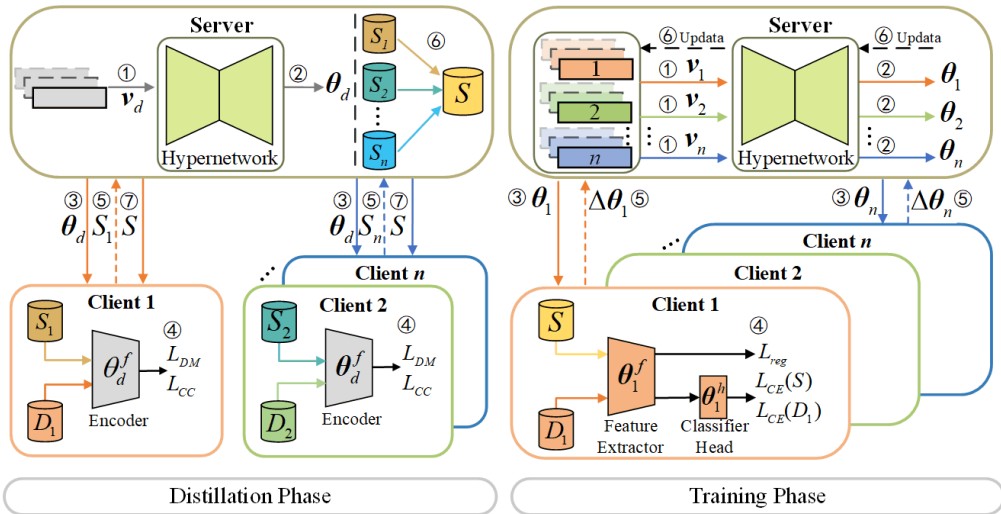

Figure 1: Framework of MH-pFedHNDD: it includes the distillation and the training Phase. **The Distillation Phase** is iterated once and contains 7 steps: ① Initialize the embedding vector $v_d$ of the distillation model and input it into the hypernetwork; ② Obtain the model parameters $\theta_d$ of the distillation model; ③ Deliver $\theta_d$ to client $i$; ④ Obtain distillation data $S_i$; ⑤ Upload $S_i$; ⑥ Aggregate $S_i$ to synthesize dataset $S$. **The Training Phase** consists of 6 steps: ① Input the embedding vector $v_i$ of client $i$ into the hypernetwork; ② Obtain personalized model parameters $\theta_i$; ③ Deliver $\theta_i$; ④ Train the local model by combining local data $D_i$ and synthetic data $S$; ⑤ Upload the model parameter update $\Delta\theta_i$; ⑥ The server obtains the gradients of the hypernetwork and the embedding vector based on $\Delta\theta_i$, thereby updating the hypernetwork and the embedding vector.

## 3 METHOD

In this section, we present our model-heterogeneous pFL method based on hyperNetworks with data distillation, MH-pFedHNDD. The overall framework is illustrated in Figure 1. This method comprises two phases: the data distillation phase and the personalized model training phase. Specifically, the distillation phase consists of 7 steps, while the personalized model training phase includes 6 steps. The details of our complete algorithm are provided in Algorithm 1.

### 3.1 PROBLEM FORMULATION

Our objective is to devise a model-heterogeneous pFL framework that tackles the challenges of both model heterogeneity and data heterogeneity. This could be formulated as a minimization problem designed to simultaneously personalize objectives for each client and adapt to heterogeneous data distributions and model architectures.

$$\{\boldsymbol{\theta}_1^*, \ldots, \boldsymbol{\theta}_n^*\} = \arg\min_{\boldsymbol{\theta}_1, \ldots, \boldsymbol{\theta}_n} \sum_{i=1}^n \mathbb{E}_{x,y \sim P_i} [\ell_i(x, y; \boldsymbol{\theta}_i)]. \tag{1}$$

Here, $P_i$ defines the data distribution specific to client $i$, while $\ell_i$ is its corresponding loss function, parameterized by the client's unique model weights $\boldsymbol{\theta}_i$. Critically, the parameter vector $\boldsymbol{\theta}_i$ and the architecture it represents can vary across the different clients, enabling the framework to handle the model heterogeneity.

To formalize the training procedure, we define the objective function as

$$\arg\min_{\boldsymbol{\theta}_1, \ldots, \boldsymbol{\theta}_n} \sum_{i=1}^n L_i(\boldsymbol{\theta}_i) = \arg\min_{\boldsymbol{\theta}_1, \ldots, \boldsymbol{\theta}_n} \sum_{i=1}^n \frac{1}{o_i} \sum_{j=1}^{o_i} \ell_i(x_j, y_j; \boldsymbol{\theta}_i). \tag{2}$$

Here, $o_i$ is the sample size of client $i$, and $L_i(\boldsymbol{\theta}_i)$ is its local empirical loss. Each client optimizes $\boldsymbol{\theta}_i$ to fit its specific data, ensuring the resulting parameters are adapted to its data distribution and task.

### 3.2 MOTIVATION

To address both model heterogeneity and data heterogeneity, we employ a hypernetwork-based approach to solve the pFL problem. However, hypernetworks typically demand substantial computational resources. To mitigate this, in Sec 3.3, we introduce a structure-sharing strategy that allows clients with similar parameter distributions to reuse model components. While this alleviates the computational burden, it still falls short in effectively balancing shared (global) and personalized (local) representations. In Sec 3.4, we take a data-driven perspective and introduce a data distillation phase. We present our distillation method and describe how synthetic data is made more compact with the aid of a real data anchor, enabling a better balance between commonality and individuality in client models. In Sec 3.5, we further describe the local training phase, where we propose techniques to improve the generalization ability of local clients. Building

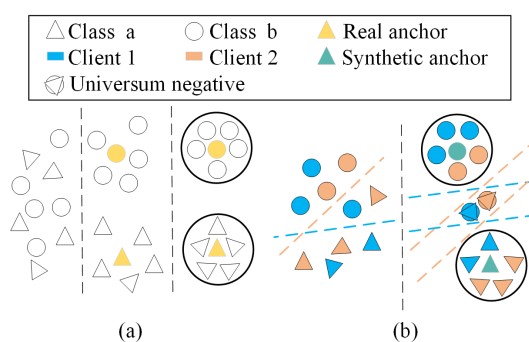

Figure 2: Figure (a): The left shows the data features generated using data distillation; the middle uses features generated with the $L_{DM}$ Loss; and the right shows features generated with the $L_{CC}$ Loss. Figure (b): Compared to the left, the right incorporates the $L_{Reg}$.

on this, we ultimately enhance the personalized performance of each local model.

### 3.3 HYPERNETWORK PERSONALIZED FEDERATED LEARNING BACKBONE

We adopt the state-of-the-art hypernetwork pFL method proposed by Zhang et al. (2025) to generate the parameters for client-specific models. Let $h(\cdot; \boldsymbol{\varphi})$ denote the hypernetwork $h$ with parameters $\boldsymbol{\varphi}$, where $\boldsymbol{\varphi}$ is composed by a feature extractor $\boldsymbol{\varphi}_f$ and multiple heads $\{\boldsymbol{\varphi}_{H_l}\}$. Clients with the same number of embedding vectors will share the same heads. For client $i$ (associated with the $l$-th head), we use $\boldsymbol{v_i} = [\boldsymbol{v}_i^1, \ldots, \boldsymbol{v}_i^{\tau_l}]$ to denotes the customized embedding vectors, where $\tau_l$ is the number of embedding vectors. Each embedding vector generates a subset of the parameters for the client model $\boldsymbol{\theta}_i$ within the hypernetwork, which is formulated as $\boldsymbol{\theta}_i^j = h(\boldsymbol{v}_i^j; \boldsymbol{\varphi}_f, \boldsymbol{\varphi}_{H_l})$. Finally, the personalized model parameters for client $i$ are generated as follows:

$$\boldsymbol{\theta}_i := \text{concat}(\boldsymbol{\theta}_i^1, \boldsymbol{\theta}_i^2, \cdots, \boldsymbol{\theta}_i^{\tau_l})_{[1:K_i]}, \tag{3}$$

where $K_i$ denotes the number of parameters of the client's personalized model, and $[1 : K_i]$ indicates that we only take the first $K_i$ parameters, ensuring that the number of generated parameters matches that of the client's personalized model. For convenience, we denote $\boldsymbol{\theta}_i = \boldsymbol{\theta}_i(\boldsymbol{\varphi}) := h(\boldsymbol{v}_i; \boldsymbol{\varphi})_{[1:K_i]}$.

### 3.4 SYNTHETIC DATASET GENERATION

Following Sec 3.3, we generate the distillation model for clients. We use $\boldsymbol{v_d} = [\boldsymbol{v}_d^1, \ldots, \boldsymbol{v}_d^{\tau_d}]$ to denotes the embedding vectors for the distilled model $\boldsymbol{\theta_d} = \boldsymbol{\theta_d}(\boldsymbol{\varphi}) := h(\boldsymbol{v_d}; \boldsymbol{\varphi})_{[1:K_d]}$, and $K_d$ represents the number of parameters of the distilled model.

During the optimization of synthetic data on the client side, we first employ the DM loss to promote alignment between the feature distributions of real and synthetic samples. Specifically, the DM loss minimizes the squared Euclidean distance between the mean feature representations of the real data and the synthetic data,

$$\mathcal{L}_{DM}(\boldsymbol{S}_i, \boldsymbol{D}_i) = ||\frac{1}{|\boldsymbol{D}_i|} \sum_{(x,y) \in \boldsymbol{D}_i} \boldsymbol{\theta}_d^f(x) - \frac{1}{|\boldsymbol{S}_i|} \sum_{(\tilde{x},\tilde{y}) \in \boldsymbol{S}_i} \boldsymbol{\theta}_d^f(\tilde{x})||^2, \tag{4}$$

where $\boldsymbol{\theta}_d^f$ denotes the feature extractor of the distillation model $\boldsymbol{\theta}_d$. Thereby encourages the synthetic data to capture the statistical characteristics and distributional tendencies of the original dataset.

However, aligning only the mean representations may not guarantee sufficient class-wise discriminability of the synthetic data. Therefore, we introduce the SupConLoss (Khosla et al., 2020) as

our Contrastive Condensation Loss ($L_{CC}$), which utilizes class labels to promote intra-class feature compactness and inter-class separability. This enhances the semantic alignment and discriminative power of the synthetic data in the feature space.

$$\mathcal{L}_{CC}(\boldsymbol{S}_i, \boldsymbol{D}_i) = \sum_{x_j \in \boldsymbol{D}_i \cup \boldsymbol{S}_i} -\frac{1}{|(\boldsymbol{D}_i \cup \boldsymbol{S}_i)_{\backslash j}^{y_j}|} \sum_{x_p \in (\boldsymbol{D}_i \cup \boldsymbol{S}_i)_{\backslash x_j}^{y_j}} \log \frac{\exp\left(\boldsymbol{\theta}_d^f(x_j) \cdot \boldsymbol{\theta}_d^f(x_p)/\tau_{\text{temp}}\right)}{\sum_{x_a \in (\boldsymbol{D}_i \cup \boldsymbol{S}_i)_{\backslash x_j}} \exp\left(\boldsymbol{\theta}_d^f(x_j) \cdot \boldsymbol{\theta}_d^f(x_a)/\tau_{\text{temp}}\right)}, \quad (5)$$

where $(\boldsymbol{D}_i \cup \boldsymbol{S}_i)_{\backslash x_j}$ represents the dataset containing both local data $\boldsymbol{D}_i$ and synthetic data $\boldsymbol{S}_i$ but without data $x_j$, $(\boldsymbol{D}_i \cup \boldsymbol{S}_i)_{\backslash x_j}^{y_j}$ a subset of $(\boldsymbol{D}_i \cup \boldsymbol{S}_i)_{\backslash x_j}$ only with samples belonging to class $y_j$, and $\tau_{temp}$ is a scalar temperature parameter.

Finally, the overall optimization objective is formulated as a weighted combination of the DM loss and the CC loss.

$$\mathcal{L}_{\text{total}}(\boldsymbol{S}_i, \boldsymbol{D}_i) = \mathcal{L}_{\text{DM}}(\boldsymbol{S}_i, \boldsymbol{D}_i) + \lambda_{\text{CC}}\mathcal{L}_{\text{CC}}(\boldsymbol{S}_i, \boldsymbol{D}_i). \quad (6)$$

Here, $\lambda_{\text{CC}}$ is a hyperparameter that balances the contributions of the two terms. In Figure 2a, we could observe that $L_{DM}$ helps synthetic features align more closely with real data features, using the latter as anchors, while $L_{CC}$ Loss encourages the synthetic features to be more compact and intra-class cohesive.

After each client $i$ completes data distillation and uploads $S_i$, the server aggregates all $S_i$ into a final synthesized dataset $S$ by merging data with the same labels. The server then sends $S$ to all clients for the next training phase.

$$S = \bigcup_{c=0}^{C-1} \left( \bigcap_{i=1}^{n} S_i^c \right). \quad (7)$$

## 3.5 PERSONALIZED LOCAL TRAINING

Following Sec 3.3, we generate the local model for clients. we use $\boldsymbol{v}_i = [\boldsymbol{v}_i^1, \ldots, \boldsymbol{v}_i^{\tau_i}]$ to denotes the customized embedding vectors for the $i$-th client $\boldsymbol{\theta}_i = \boldsymbol{\theta}_i(\boldsymbol{\varphi}) := h(\boldsymbol{v}_i; \boldsymbol{\varphi})_{[1:K_i]}$, and $K_i$ represents the number of parameters of the personalized model.

During the training phase, we optimize the model using both the local data $\boldsymbol{D}_i$ and the synthetic data $\boldsymbol{S}$. Specifically, we introduce the UniConLoss (Han et al., 2022) as our Reg Loss for feature regularization; its effect is shown in Figure 2(b).

$$\mathcal{L}_{Reg}(\boldsymbol{\theta}_i^f, \boldsymbol{S}, \boldsymbol{D}_i) = -\sum_{x_j \in \boldsymbol{B}} \log \frac{\exp\left(\boldsymbol{\theta}_i^f(x_j) \cdot \boldsymbol{m}/\tau_{\text{temp}}\right)}{\sum_{x_a \in \boldsymbol{B}_{u\backslash j}} \exp\left(\boldsymbol{\theta}_i^f(x_j) \cdot \boldsymbol{\theta}_i^f(x_a)/\tau_{\text{temp}}\right)}, \quad (8)$$

where $\boldsymbol{m} = (\sum_{x_p \in \boldsymbol{B}_{u\backslash j}^{y_i}} \boldsymbol{\theta}_i^f(x_p))/|\boldsymbol{B}_{u\backslash j}^{y_j}|$, $\boldsymbol{B}$ represents a batch containing both local data batch $\boldsymbol{B}_{\boldsymbol{D}_i}$ and synthetic data batch $\boldsymbol{B}_{\boldsymbol{S}}$. $\boldsymbol{B}_u$ represents a batch containing both the universum data batch $\boldsymbol{B}_{\boldsymbol{D}_i}^{\boldsymbol{u}}$ and the synthetic data batch $\boldsymbol{B}_{\boldsymbol{S}}$. For any $u_j \in \boldsymbol{B}_{\boldsymbol{D}_i}^{\boldsymbol{u}}$, $u_j = \lambda \cdot x_j + (1 - \lambda) \cdot x_k$ ($k \neq j$) where $x_j, x_k \in \boldsymbol{B}_{\boldsymbol{D}_i}$, and $\lambda$ denotes the mixup parameter. $\boldsymbol{B}_{u\backslash j}$ represents the subset of $\boldsymbol{B}_u$ that excludes the $j$-th data point $u_j$. $\boldsymbol{B}_{u\backslash j}^{y_i}$ a subset of $\boldsymbol{B}_{u\backslash j}$ only with samples belonging to class $y_j$. $\boldsymbol{\theta}_i^f$ denotes the feature extractor of personalized model $\boldsymbol{\theta}_i$, and $\tau_{temp}$ is the temperature hyperparameter.

Figure 2 shows that $L_{Reg}$ guides the optimization direction for generalization, which helps local clients map their local data features toward the latent embeddings of synthetic data that serve as anchors. Additionally, the introduction of universum negatives enables clients to learn larger inter-class margins, enhancing feature discriminability across different categories. Finally, the loss function is defined as follows:

$$\mathcal{L} = \mathcal{L}_{CE}(\boldsymbol{D}_i; \boldsymbol{\theta}_i) + \lambda_S \mathcal{L}_{CE}(\boldsymbol{S}; \boldsymbol{\theta}_i) + \lambda_{Reg}\mathcal{L}_{Reg}(\boldsymbol{D}_i, \boldsymbol{S}; \boldsymbol{\theta}_i). \quad (9)$$

Here, $\lambda_S$ and $\lambda_{Reg}$ are the hyperparameters.

Table 1: Accuracy comparison of FL methods (best in **bold**, second best underlined). The upper part represents homogeneous settings, while the lower part represents heterogeneous settings.

| Methods | $\alpha = 0.02$ | | | $\alpha = 0.05$ | | | $\alpha = 0.1$ | | |
|---|---|---|---|---|---|---|---|---|---|
| | CIFAR-10 | CIFAR-100 | Tiny-ImageNet | CIFAR-10 | CIFAR-100 | Tiny-ImageNet | CIFAR-10 | CIFAR-100 | Tiny-ImageNet |
| FedAvg (McMahan et al., 2017) | 47.45 | 22.34 | 11.46 | 59.80 | 24.86 | 12.44 | 61.76 | 31.70 | 13.25 |
| FedBN (Li et al., 2021) | 82.64 | 25.54 | 19.67 | 82.59 | 25.41 | 15.79 | 78.27 | 29.08 | 14.71 |
| FedDM (Xiong et al., 2023) | 65.95 | 50.40 | 26.23 | 67.53 | 51.13 | 31.82 | 67.30 | 57.73 | 39.95 |
| FedAF (Wang et al., 2024b) | 47.20 | 39.65 | 25.16 | 53.48 | 42.71 | 29.74 | 56.29 | 47.00 | 34.47 |
| pFedHN (Shamsian et al., 2021) | 94.20 | 65.55 | 36.74 | 90.34 | 61.73 | 35.96 | 85.82 | 57.93 | 27.51 |
| pFedLHN (Zhu et al., 2023) | 95.60 | 72.68 | 49.13 | 92.22 | 67.92 | 44.65 | 88.12 | 63.60 | 38.37 |
| MH-pFedHN (Zhang et al., 2025) | 95.81 | 75.53 | 50.14 | 93.15 | 71.07 | 47.28 | 89.23 | 66.89 | 40.23 |
| MH-pFedHNGD (Zhang et al., 2025) | 96.26 | 76.41 | 47.54 | 93.09 | 72.30 | 42.86 | 89.23 | **68.12** | 36.79 |
| MH-pFedHNDD | **96.81** | **77.00** | **51.46** | **94.05** | **72.58** | **47.84** | **90.13** | 68.02 | **41.05** |
| *Difference* | 0.55 | 0.59 | 1.32 | 0.90 | 0.28 | 0.56 | 0.90 | -0.1 | 0.79 |
| FedGH (Yi et al., 2023a) | **93.91** | 52.50 | 31.87 | **88.56** | 55.39 | 28.61 | **84.32** | 45.40 | 28.07 |
| DESA (Huang et al., 2024) | 92.30 | 58.21 | 36.34 | 84.03 | 53.39 | 30.42 | 78.69 | 46.12 | 25.40 |
| pFedHN (Shamsian et al., 2021) | 91.34 | 58.64 | 32.37 | 83.95 | 53.50 | 25.27 | 79.02 | 47.05 | 19.52 |
| pFedLHN (Zhu et al., 2023) | 91.80 | 60.76 | 36.84 | 84.34 | 55.44 | 33.03 | 79.21 | 50.18 | 26.21 |
| FedTGP (Zhang et al., 2024b) | 92.66 | 56.28 | 29.16 | 87.24 | 50.80 | 24.50 | 81.85 | 41.95 | 16.60 |
| MH-pFedHN (Zhang et al., 2025) | 92.12 | 61.99 | 39.30 | 84.84 | 56.91 | 35.47 | 79.89 | 52.00 | 29.39 |
| MH-pFedHNGD (Zhang et al., 2025) | 90.74 | 64.39 | 40.34 | 86.07 | 58.32 | 36.23 | 80.42 | **55.70** | 30.02 |
| MH-pFedHNDD | 91.75 | **65.02** | **40.88** | 85.35 | **59.50** | **36.45** | 80.06 | 54.80 | **30.44** |
| *Difference* | -2.16 | 0.63 | 0.54 | -3.21 | 1.18 | 0.22 | -4.26 | -0.9 | 0.42 |

## 4 EXPERIMENTS

**Datasets.** We evaluate our approach and baselines on three popular image classification datasets, including CIFAR-10, CIFAR-100 (Krizhevsky & Hinton, 2009), and Tiny-ImageNet (Le & Yang, 2015). We adopt two non-IID settings (T Dinh et al., 2020; Lin et al., 2020; Li et al., 2022; Liu et al., 2024). 1) Distribution-based label imbalance. We employ the Dirichlet distribution $Dir(0.02)$, $Dir(0.05)$, and $Dir(0.1)$ to partition the dataset among the clients. 2) Quantity-based label imbalance. For the CIFAR-10, CIFAR-100, and Tiny-ImageNet datasets, we randomly allocate 2, 10, and 20 classes to each client, respectively. We draw $\alpha_{i,c} \sim U(0.4, 0.6)$, and allocate $\frac{\alpha_{i,c}}{\sum_j \alpha_{j,c}}$ of the samples for the class $c$ selected on client $i$. For each client, 75% of the data is used for training, and 25% is used for testing.

**Baselines.** We choose various state-of-the-art methods. For federated data distillation methods, we adopt FedDM (Xiong et al., 2023) and FedAF (Wang et al., 2024b); for personalized federated learning, we consider FedGH (Yi et al., 2023a) and FedTGP (Zhang et al., 2024b); for hypernetwork-based approaches, we evaluate pFedHN (Shamsian et al., 2021), pFedLHN (Zhu et al., 2023), MH-pFedHN and MH-pFedHNGD (Zhang et al., 2025); and for methods targeting the joint achievement of commonality and personalization, we include DESA (Huang et al., 2024). In addition, we also incorporate **FedAvg** (McMahan et al., 2017) and **FedBN** (Li et al., 2021). In this way, we are able to validate the effectiveness of our method from multiple perspectives.

**Model heterogeneity.** In prior studies on federated dataset distillation, each client employs ConvNet (Zhao et al., 2020b). For a fair comparison, we use ConvNet in homogeneous model experiments. In addition, we also use MLP, LeNet (LeCun et al., 1998), VGGNet (Simonyan & Zisserman, 2015) and ResNet-9 (He et al., 2016) in heterogeneous model settings. All our heterogeneous experiments use these five models, which are evenly distributed among all clients by default.

**Training Strategies.** We configure FedAvg, FedBN, FedGH, and FedTGP with 200 communication rounds and 10 local epochs per round. For FedDM and FedAF, we adopt 20 communication rounds, a total of 500 global training epochs, and 1000 local distillation iterations, using 50 images per class (IPC) for synthetic data generation. For DESA, we set 1000 local distillation iterations, 100 local training epochs, and use IPC = 50. For pFedHN, pFedLHN, MH-pFedHN, and MH-pFedHNGD, we use 500 communication rounds with 1 local epoch per round. Specifically, for MH-pFedHNGD, we adopt ConvNet as the global model. For MH-pFedHNDD, we set 500 communication rounds with 1 local epoch, and 3000 local distillation iterations. IPC values are set to 50 for CIFAR-10, 10 for CIFAR-100, and 10 for Tiny-ImageNet, respectively. The default client number is 10. For hyperparameter, we set $\lambda = 0.7$, $\lambda_{CC} = 5e-4$, $\lambda_S = 0.1$ and $\lambda_{Reg} = 0.1$. All results are averaged across **five** random seeds. More settings and design choices are in Appendix B and Appendix D.

### 4.1 AN OVERALL COMPARISON

We compare the accuracy of MH-pFedHNDD against other baselines in Table 1 and Table 2. The experimental results demonstrate that under both non-IID settings and across homogeneous and het-

Table 2: Accuracy comparison of FL methods with quantity-based label imbalance. The upper part represents homogeneous settings, while the lower part represents heterogeneous settings.

| Methods | CIFAR-10 | CIFAR-100 | Tiny-ImageNet |
|---|---|---|---|
| FedAvg (McMahan et al., 2017) | 53.10 | 28.22 | 17.66 |
| FedBN (Li et al., 2021) | 87.16 | 35.21 | 24.55 |
| FedDM (Xiong et al., 2023) | 71.69 | 53.88 | 34.36 |
| FedAF (Wang et al., 2024b) | 53.85 | 43.49 | 30.43 |
| pFedHN (Shamsian et al., 2021) | 94.45 | 69.36 | 40.45 |
| pFedLHN (Zhu et al., 2023) | 95.59 | 75.60 | 52.17 |
| MH-pFedHN (Zhang et al., 2025) | 96.61 | 78.48 | 55.43 |
| MH-pFedHNGD (Zhang et al., 2025) | 96.89 | 79.24 | 48.67 |
| MH-pFedHNDD | **97.33** | **79.79** | **56.00** |
| *Difference* | 0.44 | 0.55 | 0.57 |
| FedGH (Yi et al., 2023a) | 92.70 | 63.82 | 39.23 |
| DESA (Huang et al., 2024) | 92.55 | 62.86 | 38.76 |
| pFedHN (Shamsian et al., 2021) | 92.62 | 62.58 | 37.95 |
| pFedLHN (Zhu et al., 2023) | 93.16 | 64.82 | 40.78 |
| FedTGP (Zhang et al., 2024b) | 92.52 | 60.54 | 37.59 |
| MH-pFedHN (Zhang et al., 2025) | 93.36 | 66.76 | 42.79 |
| MH-pFedHNGD (Zhang et al., 2025) | **94.38** | 69.02 | 44.57 |
| MH-pFedHNDD | 92.34 | **69.26** | **45.62** |
| *Difference* | -2.04 | 0.24 | 1.05 |

Table 3: Impact of core design on the personalized model accuracy for learning CIFAR-100 under three different degrees of heterogeneity.

| Configuration | $\alpha = 0.02$ | $\alpha = 0.05$ | $\alpha = 0.1$ |
|---|---|---|---|
| MH-pFedHNDD w/o $L_{CC}$ | 76.82 | 72.19 | 67.54 |
| MH-pFedHNDD w/o UniNeg | 76.87 | 72.55 | 67.79 |
| MH-pFedHNDD | 77.00 | 72.58 | 68.02 |

Table 4: Impact of IPC on the personalized model accuracy for learning CIFAR-100 under three different degrees of heterogeneity.

| Configuration | $\alpha = 0.02$ | $\alpha = 0.05$ | $\alpha = 0.1$ |
|---|---|---|---|
| IPC=2 | 74.96 | 71.16 | 66.57 |
| IPC=5 | 75.88 | 72.02 | 67.30 |
| IPC=10 | 77.00 | 72.58 | 68.02 |
| IPC=25 | 77.50 | 72.80 | 68.74 |
| IPC=50 | 76.92 | 72.88 | 68.67 |

erogeneous model scenarios, our method consistently achieves the best or near-best performance. In the distribution-based label imbalance setting on CIFAR-10, the simplicity of the dataset may limit the benefits of synthetic data generation, causing MH-pFedHNDD to perform slightly below the state of the art. Similarly, in the more challenging quantity-based label imbalance setting, the model-driven approach MH-pFedHNGD achieves comparable results to MH-pFedHNDD on CIFAR-10. On CIFAR-100 and Tiny-ImageNet, MH-pFedHNDD achieves significantly better performance, highlighting its effectiveness in addressing the more challenging problems of complex datasets in pFL from a data-driven perspective. Moreover, it demonstrates a better balance between personalization and generalization.

## 4.2 EXPERIMENT WITH GENERALIZATION

In federated learning, generalization measures how well a method performs on unseen clients. We conduct experiments on CIFAR-100, where 80% of the clients are used to train the hyper-network, and the remaining 20% are held out for generalization testing. During testing, the hypernetwork parameters are fixed.

Figure 3 presents results under both homogeneous and heterogeneous settings, across different levels of non-IID settings. Notably, the state-of-the-art MH-pFedHN is only comparable to MH-pFedHNDD when $\alpha = 0.02$ in the homogeneous setting. In all other scenarios, MH-pFedHNDD consistently outperforms MH-pFedHN. These results demonstrate that our design is more effective in improving the generalization ability of the hypernetwork, which in turn enhances local clients' personalized performance.

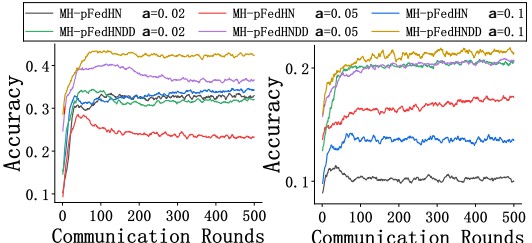

Figure 3: Generalization experiments, the left denotes homogeneous settings and the right denotes heterogeneous settings.

## 4.3 ABLATION STUDY

Table 3 presents the impact of different components of our framework. Without $L_{CC}$, the accuracy drops more severely as the data distribution scale increases, ranging from 0.18% to 0.48%. A similar trend is observed when the universum negatives are removed. These results demonstrate that $L_{CC}$ indeed helps synthetic data become more compactly anchored to real data, thereby producing more precise synthetic datasets and improving performance. Meanwhile, UniNeg enables clients to form larger inter-class decision boundaries, enhancing personalized performance.

Table 5: Performance improvement on CIFAR-100 by integrating data distillation into other federated learning methods.

| Methods | $\alpha = 0.02$ | | $\alpha = 0.05$ | | $\alpha = 0.1$ | |
|---|---|---|---|---|---|---|
| | Acc | Diff | Acc | Diff | Acc | Diff |
| FedAvg | 35.48 | 13.14 | 35.67 | 10.81 | 37.55 | 5.85 |
| FedBN | 38.04 | 12.50 | 38.18 | 12.77 | 38.71 | 9.63 |
| pFedHN | 67.31 | 1.76 | 62.18 | 0.45 | 58.02 | 0.09 |
| pFedLHN | 75.84 | 3.16 | 71.55 | 3.63 | 66.13 | 2.53 |
| FedGH | 61.22 | 8.72 | 56.95 | 1.56 | 50.69 | 5.29 |
| pFedHN | 59.43 | 0.79 | 53.82 | 0.32 | 48.61 | 1.56 |
| pFedLHN | 63.29 | 2.53 | 58.04 | 2.60 | 52.34 | 2.16 |

Table 6: The results for MH-pFedHNDD trained with Different Privacy (DP).

| Configuration | $\alpha = 0.02$ | $\alpha = 0.05$ | $\alpha = 0.1$ |
|---|---|---|---|
| MH-pFedHNDD | 77.00 | 72.58 | 68.02 |
| MH-pFedHNDD(DP) | 75.70 | 71.85 | 67.02 |

Table 7: Experiments with scalability, upper (homogeneous) and lower (heterogeneous).

| Client Number | $\alpha = 0.02$ | $\alpha = 0.05$ | $\alpha = 0.1$ |
|---|---|---|---|
| 50 | 80.33 | 73.01 | 66.67 |
| 100 | 79.33 | 71.41 | 63.88 |
| 200 | 80.15 | 68.71 | 61.28 |
| 50 | 82.41 | 75.64 | 68.84 |
| 100 | 81.84 | 73.28 | 65.75 |
| 200 | 82.27 | 70.34 | 63.41 |

## 4.4 EXPERIMENTS WITH IMPACT OF IPC

We conduct IPC experiments on the CIFAR-100 dataset, where the number of IPC is varied from 2 to 50. Table 4 shows that as IPC increases, the accuracy also improves. However, when IPC is set to 50, the accuracy slightly drops. This is because each client generates 5,000 synthetic images for CIFAR-100, which reduces the quality of some synthetic data and lowers the relative proportion of real data during training, ultimately causing the accuracy to decline. Nevertheless, the accuracy at IPC=50 remains higher than at IPC=2 or 5, demonstrating the robustness of our MH-pFedHNDD.

## 4.5 EFFICIENT PLUG-AND-PLAY PROPERTY

Table 5 reports the results of integrating our data distillation method into different baselines. The upper part presents the results under the homogeneous setting, while the lower part shows the results under the heterogeneous setting. All methods benefit from an accuracy improvement of up to 13.14%. Although the results still fall short compared with MH-pFedHNDD, these experiments demonstrate the potential of addressing the pFL problem from a data-driven perspective.

## 4.6 EXPERIMENTS WITH PRIVACY PRESERVATION

We conducted differential privacy (Abadi et al., 2016) experiments for MH-pFedHNDD when generating synthetic data. Table 6 shows that MH-pFedHNDD can be effectively combined with differential privacy (DP) while still maintaining high accuracy, with only about a 1% drop. The results remain comparable to the state-of-the-art, demonstrating that our method can be seamlessly integrated with privacy-preserving techniques and still achieve strong performance.

## 4.7 EXPERIMENTS WITH SCALABILITY

Table 7 presents the experimental results of our method with the number of clients set to 50, 100, and 200. We observe that as the degree of non-IID increases, the accuracy usually decreases with a larger number of clients. Nevertheless, our method still achieves strong performance—for instance, under the heterogeneous setting with 200 clients, the accuracy remains as high as 82.27. These results demonstrate the scalability of MH-pFedHNDD.

## 5 CONCLUSION

In this paper, we are the first to propose MH-pFedHNDD from a data-driven perspective, leveraging data distillation within hypernetworks to address the challenges of model heterogeneity and data heterogeneity in pFL. By introducing the Contrastive Condensation Loss and the Reg loss, we enable the generation of more accurate synthetic data and enhance the generality of local clients. Furthermore, MH-pFedHNDD incorporates universum negatives to ultimately improve personalized performance. We conduct extensive experiments against a variety of baselines, which validate the effectiveness of our method and highlight the promising potential of data-driven approaches.

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

# Part I

# Appendix

## Table of Contents

## A  ALGORITHMS

Here, we outline the pseudo-algorithm of MH-pFedHNDD in Algorithm 1.

## B  ADDITIONAL EXPERIMENT SETTINGS

### B.1  DATASET DETAILS

We use three widely adopted datasets to evaluate our proposed methods. Both CIFAR-10 and CIFAR-100 (Krizhevsky & Hinton, 2009) consist of 60,000 color images, each with dimensions of $32 \times 32$ pixels. CIFAR-10 contains 10 different classes, with each class containing 6000 images. CIFAR-100 includes 100 different classes, with each class containing 600 images. Tiny-ImageNet (Le & Yang, 2015) comprises 100,000 color images with dimensions of $64 \times 64$ pixels each. It includes 200 classes, with 500 images per class.

### B.2  EXPERIMENT SETTINGS

For traditional federated learning methods (FedAvg and FedBN), we set the learning rate to 0.01. For federated data distillation methods such as FedDM and FedAF, we sample data from the real dataset. The distillation learning rate is set to 1.0, while the federated learning rate is 0.01. For personalized federated learning baselines, we use a learning rate of 5e-3 for FedGH and 0.01 for FedTGP. The learning rate for DESA is set to 0.01. For pFedHN, pFedLHN, MH-pFedHN, and MH-pFedHNGD, the learning rates for the hypernetworks are set to 1e-3, 3e-4, 2e-4, and 2e-4, respectively. For MH-pFedHNDD (our proposed method), the hypernetwork learning rate is set to 2e-4. All of these methods use a local client learning rate of 1e-3 with a weight decay of 1e-4.

---

**Algorithm 1** Model-Heterogeneous Personalized Federated Hypernetwork with Dataset Distillation

---

**Input:** $R$ - number of rounds; $\alpha, \beta, \gamma, \eta$ - learning rate; $E$ - client local epoch; $T$ - number of iterations for dataset distillation; $C$ - Number of classes; $\{K_1, \ldots, K_n\}$ - number of clients parameter; $\{D_1, \ldots, D_n\}$ - datasets; $\lambda_S$ - weight of distillation dataset loss; $\lambda_{Reg}$ - weight of regularization term.

**Output:** trained model parameters $\{\theta_1, \ldots, \theta_n\}$

  **procedure** SERVER EXECUTES
    $\theta_d = h(v_d; \varphi)_{[1:K_d]}$
    **for** each client $i$ **in parallel do**
      $S_i \leftarrow$ ClientDatasetDistillation($\theta_d$)
    **end for**
    get $S$ based on Equation 7, and send it to all clients
    **for** $r = 1$ **to** $R$ **do**
      **for** each client $i$ **in parallel do**
        $\theta_i = h(v_i; \varphi)_{[1:K_i]}$
        $\Delta\theta_i \leftarrow$ ClientUpdate($\theta_i$)
        $\varphi = \varphi - \alpha\nabla_\varphi\theta_i^T\Delta\theta_i$
        $v_i = v_i - \alpha\nabla_{v_i}\varphi^T\nabla_\varphi\theta_i^T\Delta\theta_i$
      **end for**
    **end for**
  **end procedure**
  **function** CLIENTDATASETDISTILLATION($\theta_d$)
    initialize $S_i$ with a subset of $D_i$
    **for** $t = 1$ **to** $T$ **do**
      sample batch $B_{D_i} \subset D_i$
      **for** $c = 1$ **to** $C$ **do**
        sample batch $B_{D_i}^c \subset B_{D_i}$, $S_i^c \subset S_i$
        $S_i^c = S_i^c - \beta\nabla_{\theta_d}L_{DM}(\theta_d, B_{D_i}^c, S_i^c)$
      **end for**
      $S_i = S_i - \gamma\nabla_{\theta_d}L_{CC}(\theta_d, B_{D_i}, S_i)$
    **end for**
    **return** $S_i$
  **end function**
  **function** CLIENTUPDATE($\theta_i$)
    $\widetilde{\theta}_i = \theta_i$
    **for** $e = 1$ **to** $E$ **do**
      sample batch $B_{D_i} \subset D_i$, $B_S \subset S$
      $\widetilde{\theta}_i = \widetilde{\theta}_i - \eta\nabla_{\widetilde{\theta}_i}\left(L(\widetilde{\theta}_i, B_{D_i}) + \lambda_S L(\widetilde{\theta}_i, B_S) + \lambda_{Reg}L(\widetilde{\theta}_i, B_{D_i}, B_S)\right)$
    **end for**
    $\Delta\theta_i = \widetilde{\theta}_i - \theta_i$
    **return** $\Delta\theta_i$
  **end function**

---

### B.3 IMPLEMENTATION

We conduct all the experiments on a workstation with a 2.6-GHz Intel W7-2475X CPU, an RTX 4090 GPU and 125 GiB of RAM. All the code is written using PyTorch.

## C ADDITIONAL EXPERIMENTS

### C.1 EXPERIMENTS WITH DIFFERENT CLIENT PARTICIPATION RATIOS

In Table 8, we report the results obtained by varying the client participation ratios. As expected, higher participation ratios lead to higher accuracy. Nevertheless, even in the most biased case with $\alpha = 0.1$ and a ratio of 0.2, the accuracy still exceeds 66%, demonstrating the robustness of our method to different participation ratios.

Table 8: MH-pFedHNDD experiments with different participation ratios.

| Ratio | $\alpha = 0.02$ | $\alpha = 0.05$ | $\alpha = 0.1$ |
|---|---|---|---|
| 0.2 | 74.51 | 70.9 | 66.42 |
| 0.4 | 76.06 | 72.27 | 67.82 |
| 0.6 | 76.18 | 73.39 | 67.87 |
| 0.8 | 76.47 | 72.72 | 67.98 |
| 1.0 | 77.00 | 72.58 | 68.02 |

Table 9: Experiments with different sources of data.

| Configuration | $\alpha = 0.02$ | $\alpha = 0.05$ | $\alpha = 0.1$ |
|---|---|---|---|
| MH-pFedHN | 75.53 | 71.07 | 66.89 |
| Only $S$ | 24.05 | 22.60 | 22.51 |
| One-shot | 23.50 | 21.33 | 20.68 |
| MH-pFedHNDD | 77.00 | 72.58 | 68.02 |

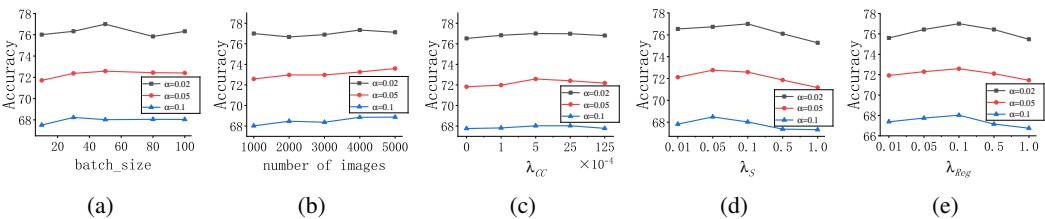

(a)  (b)  (c)  (d)  (e)

Figure 4: Figure (a): the impact of synthetic data's batch size; Figure (b): synthetic data quantity, Figure (c): $L_{CC}$ loss weight; Figure (d): synthetic data loss weight; Figure (e): $L_{Reg}$ loss weight.

### C.2 EXPERIMENTS WITH DIFFERENT SOURCES OF DATA

In Table 9, MH-pFedHN refers to a hypernetwork-based method where each client is trained only on its local dataset; Only $S$ denotes the method where clients are trained solely on the synthetic dataset; One-shot represents the setting in which each client uploads the synthetic dataset only once and the server trains a global model on it; and MH-pFedHNDD leverages both local datasets and synthetic dataset. The experimental results demonstrate that local data play a decisive role in pFL. Nevertheless, training solely on synthetic data can still yield effective results, with improvements of up to 24.05%. Combining both local datasets and synthetic data leads to even better performance, highlighting the effectiveness of our data-driven approach.

## D EXPERIMENTS WITH HYPERPARAMETERS

We use the CIFAR-100 dataset under a homogeneous setup to explore various hyperparameter configurations.

### D.1 EXPERIMENT WITH SYNTHETIC DATASET BATCH

Here, we explore the impact of the batch size of synthetic data during training on model accuracy, with results shown in Figure 4a. We found that the best performance occurs when the batch size of synthetic data is close to but slightly smaller than that of the local data (in the experiment, we set them to 50 and 64, respectively). This is because when the batch size of synthetic data is too small, the optimization of the personalized model is dominated entirely by the local data, and the synthetic data provides no benefit; conversely, when the batch size is too large, it interferes with the optimization of the model by the local data.

## D.2 Experiment with Size of the Synthetic Dataset

Here, we explored the impact of the quantity of synthetic datasets on model accuracy. The results are shown in Figure 4b. For example, when IPC=10, we need to run the data distillation process twice to generate 2,000 synthetic images, which differs from the IPC=20 setting, where 2,000 images can be generated in a single run. This is because a smaller IPC value allows each synthetic image to contain richer information from the original local data.

It can be observed that as the amount of synthetic data increases, the model accuracy continuously improves. This is because more diverse data increases the variety and coverage of training samples, thereby helping the model learn more generalized knowledge and enhancing performance.

Table 10: ConvNet model structure.

| Layer | Shape | Nonlinearity |
|---|---|---|
| Conv1 | $3 \times 3 \times 3 \times 128$ | ReLU |
| Avg-Pool | $2 \times 2$ | - |
| Conv2 | $128 \times 3 \times 3 \times 128$ | ReLU |
| Avg-Pool | $2 \times 2$ | - |
| Conv3 | $128 \times 3 \times 3 \times 128$ | ReLU |
| Avg-Pool | $2 \times 2$ | Flatten |
| FC | $2048 \times 100$ | - |

Table 11: LeNet-style model structure.

| Layer | Shape | Nonlinearity |
|---|---|---|
| Conv1 | $3 \times 3 \times 3 \times 16$ | ReLU |
| MaxPool | $2 \times 2$ | - |
| Conv2 | $16 \times 3 \times 3 \times 32$ | ReLU |
| MaxPool | $2 \times 2$ | Flatten |
| FC1 | $2048 \times 108$ | ReLU |
| FC2 | $108 \times 2048$ | ReLU |
| FC3 | $2048 \times 100$ | None |

Table 12: MLP model structure.

| Layer | Shape | Nonlinearity |
|---|---|---|
| FC1 | $3072 \times 128$ | ReLU |
| FC2 | $128 \times 2048$ | ReLU |
| FC3 | $2048 \times 100$ | None |

Table 13: Simplified VGG8 model structure.

| Layer | Shape | Nonlinearity |
|---|---|---|
| Conv1 | $3 \times 3 \times 3 \times 16$ | ReLU |
| Conv2 | $16 \times 3 \times 3 \times 16$ | ReLU |
| MaxPool | $2 \times 2$ | - |
| Conv3 | $16 \times 3 \times 3 \times 32$ | ReLU |
| Conv4 | $32 \times 3 \times 3 \times 32$ | ReLU |
| MaxPool | $2 \times 2$ | - |
| Conv5 | $32 \times 3 \times 3 \times 64$ | ReLU |
| Conv6 | $64 \times 3 \times 3 \times 64$ | ReLU |
| MaxPool | $2 \times 2$ | Flatten |
| Linear1 | $1024 \times 180$ | ReLU |
| Linear2 | $180 \times 2048$ | ReLU |
| Linear3 | $2048 \times 100$ | None |

Table 14: Structure of the 9-layer Residual network model.

| Group Name | Output Size | 9-layer ResNet |
|---|---|---|
| Conv1 | $32 \times 32$ | $[3 \times 3, 32]$ |
| Conv2 | $32 \times 32$ | $\begin{bmatrix} 3 \times 3, 32 \\ 3 \times 3, 32 \end{bmatrix} \times 3$ |
| Conv3 | $16 \times 16$ | $\begin{bmatrix} 3 \times 3, 64 \\ 3 \times 3, 64 \end{bmatrix} \times 3$ |
| Conv4 | $8 \times 8$ | $\begin{bmatrix} 3 \times 3, 128 \\ 3 \times 3, 128 \end{bmatrix} \times 3$ |
| Avg-Pool | $4 \times 4$ | $[4 \times 4]$ |

## D.3 Experiment with Loss Weights

Figure 4c illustrates the effect of varying the weight $\lambda_{CC}$ of the LCC loss. We observe that increasing $\lambda_{CC}$ from a small value helps the synthetic data become more compactly anchored to the real data, thereby improving the quality of the generated data and leading to higher accuracy. However, when $\lambda_{CC}$ becomes too large, it reduces the relative weight of the data distillation loss in Equation 6, which in turn degrades the quality of the synthetic data. The optimal hyperparameter value of $\lambda_{CC}$ is found to be around 5e-4, which is also used as the default setting in our experiments.

Figure 4d shows the effect of varying $\lambda_S$. As $\lambda_S$ increases, local training relies more on the synthetic dataset, which strengthens the data-driven perspective in optimizing the hypernetwork and thus improves accuracy. However, in Equation 9, when $\lambda_S$ becomes too large, the contribution of the real dataset to optimization diminishes, weakening personalization. At the same time, the Reg Loss can

no longer play its role effectively, reducing the generalization ability of local clients. The optimal hyperparameter for $\lambda_S$ is 0.1, which is also used as the default setting in our experiments.

Figure 4e illustrates the effect of varying $\lambda_{Reg}$. The results show that as $\lambda_{Reg}$ increases, it better guides local clients to integrate the latent embeddings of all clients' synthetic data as anchors, thereby steering the optimization toward improved generalization and performance. However, as shown in Equation 9, when $\lambda_{Reg}$ becomes too large, local training struggles to leverage the features of both local datasets and synthetic data to optimize the local models, ultimately leading to degraded performance. The optimal hyperparameter for $\lambda_{Reg}$ is also 0.1, which is used as the default setting in our experiments.

# E  MODEL ARCHITECTURES AND PARAMETERS

Here, we present all the model architectures and the parameters used in the CIFAR-100 experiments (the only difference across datasets lies in the final output layer). Table 10 is a ConvNet model, Table 11 is a LeNet-style model, Table 12 is an MLP model, Table 13 is a simplified VGG model (8 layers), Table 14 is a residual network (9 layers).

# F  THE USE OF LARGE LANGUAGE MODELS

We only use Large Language Models (LLMs) to aid or polish writing and check typos.

