# OpenReview forum: "Exploring Hypernetwork to Enhance Model Heterogeneous Personalized Federated Learning with Data Distillation"
_ICLR.cc/2026/Conference — ICLR 2026 Conference Withdrawn Submission_

### Official Review · Reviewer_QcMv · 2025-10-16

**Soundness:** 2
**Presentation:** 2
**Contribution:** 2
**Rating:** 2
**Confidence:** 4

**Summary:**

This paper introduces MH-pFedHNDD, a novel personalized federated learning framework that integrates hypernetworks with data distillation to handle both model and data heterogeneity. Each client generates synthetic data locally, which the server aggregates to enhance global knowledge without sharing raw data. Experiments on multiple datasets show that MH-pFedHNDD outperforms state-of-the-art baselines, achieving better generalization and privacy preservation.

**Strengths:**

- First work to combine hypernetworks with data distillation in personalized federated learning.
- The work provides extensive evaluation and shows good empirical results.

**Weaknesses:**

- The novelty of the paper is limited. The method primarily integrates several existing ideas, namely hypernetworks for personalized model generation (as in pFedHN, pFedLHN, MH-pFedHN) and data distillation approaches (like FedDM, FedAF), into a single framework. Its main originality lies in combining these components and adding two tailored losses (which are also fairly incremental adaptations IMHO), but the underlying techniques themselves are extensions of established methods rather than entirely new concepts.
- The framework introduces additional computational and communicational overhead (from data distillation). This may hinder scalability in resource-limited edge devices.
- Following the previous point, the experimental section lacks any results or analysis on the computational/communicational overhead. Adding such analysis would greatly benefit the work and allow for assessing its scalability.
- The paper lacks empirical/theoretical motivation and analysis explaining which components of the method contribute to handling heterogeneity and how they do so. For example, it does not show how data distillation mitigates inter-client differences, or why the proposed losses specifically improve performance in heterogeneous settings. Furthermore, from Tables 1 and 2, it appears that the improvements are more significant in the homogeneous setups, but this is not thoroughly discussed.
- Could you explain the choice of hyperparameters (e.g., communication rounds, lr etc.). From the HP section in the Appendix it doesn’t seem you applied any HP search, so how do you ensure fair comparison?
- Please report standard errors or some other measure of variability for the results. The improvements in Table 3 and other places appears very moderate, so e.g. standard errors will help asses the improvement.
- Some parts, like the "Problem Formulation", appear too similar to the relevant part in _Hypernetworks for Model-Heterogeneous Personalized Federated Learning_ (Zhang et al, 2025), with only minor modifications.
- Minor:
    - Line 112: “DM” is not yet defined.
    - Hyperlinks to ref “Huang et al., 2024” are not working.

**Questions:**

- Please explain the merging in Eq. 7.
- How does the computational and communication cost of MH-pFedHNDD compare to other hypernetwork-based methods in practice?
- Could the proposed framework extend to non-vision domains (e.g., NLP, healthcare) and other tasks (non-classification tasks)?
- While the main focus and motivation is around the heterogeneous setup, from Tables 1 and 2 it seems like the improvements are more significant in the homogeneous setups. Why is that?
- Following the previous point, the motivation for why this approach is especially effective under model and data heterogeneity is unclear. Could you please elaborate on that?
- It would be interesting and beneficial to show what the synthetic samples look like or how their distribution compares to real data.

---

### Official Review · Reviewer_bNfF · 2025-10-30

**Soundness:** 3
**Presentation:** 2
**Contribution:** 3
**Rating:** 4
**Confidence:** 3

**Summary:**

This paper investigates personalized federated learning under both data and model heterogeneity, aiming to preserve global knowledge while generating personalized models tailored to each client’s specific data distribution and model architecture. In highly heterogeneous scenarios, existing approaches often struggle to effectively capture and utilize global information, leading to a pronounced trade-off between personalization and generalization. To address this issue, the authors propose MH-pFedHNDD, which for the first time integrates data distillation into a hypernetwork-based federated learning framework, providing a data-driven perspective to mitigate this challenge. The proposed method consists of two main components: synthetic data generation and personalized local model training. Extensive experiments across multiple datasets and varying degrees of heterogeneity under both homogeneous and heterogeneous model settings demonstrate the effectiveness of the proposed approach compared with various SOTA baselines.

**Strengths:**

- Clear and meaningful problem definition. Personalized federated learning under model heterogeneity is a frontier topic, and exploring collaboration beyond parameter sharing is valuable.

- Comprehensive experimental design. Covers various Non-IID settings, comparisons with multiple baselines, and analyses on ablation, IPC, DP, and scalability.

**Weaknesses:**

- The clarity of presentation can be improved. There are minor typos and formatting issues, such as the caption of Fig.2 (“the middle uses” should be “the middle shows”), and inconsistent font sizes and spacing in tables that affect readability.

- Limited performance gain. The ablation study shows a maximum improvement of only 0.48%, which is not very significant.

**Questions:**

- As shown in Table 1, MH-pFedHNDD demonstrates stronger competitiveness under homogeneous settings, but nearly half of the results under heterogeneous settings are inferior to the SOTA. Since the paper mainly targets model heterogeneity, this inconsistency with the main objective should be further explained.
- It would be helpful if the authors could report the computational cost (in GFLOPs) of both the synthetic data generation and the hypernetwork training to better assess the method’s efficiency.
- Regarding model heterogeneity, the experiments only involve architectures such as MLP, LeNet, VGG, and ResNet-9. Do these differences sufficiently reflect the essence of model heterogeneity? Moreover, can MH-pFedHNDD maintain stable performance in more extreme heterogeneous settings, e.g., with both CNNs and Transformers?

---

### Official Review · Reviewer_S4AY · 2025-10-31

**Soundness:** 2
**Presentation:** 3
**Contribution:** 2
**Rating:** 4
**Confidence:** 3

**Summary:**

This paper focuses on Personalized Federated Learning (pFL) scenarios characterized by both model heterogeneity (clients use diverse architectures like ConvNet, MLP, and ResNet-9 due to varying computational resources) and data heterogeneity (non-IID data via Dirichlet distribution or quantity-based label imbalance). The core framework, MH-pFedHNDD, leverages a server-maintained hypernetwork to generate client-specific model parameters. Key steps include: 1) Clients use their locally generated models (from the hypernetwork) and local data to distill synthetic datasets; 2) The server aggregates all clients’ synthetic data into a global synthetic dataset and distributes it back to clients; 3) Clients train their personalized models using both local data and the global synthetic data, then upload parameter updates to the server; 4) The server optimizes the hypernetwork parameters and clients’ customized embedding vectors to refine future personalized model generation. The method aims to balance generalization (via global synthetic data) and personalization (via local data) while accommodating model heterogeneity.

**Strengths:**

Minimal scenario assumptions: MH-pFedHNDD operates effectively under realistic constraints—supporting heterogeneous computing resources (clients use diverse architectures like MLP, LeNet, ResNet-9) and heterogeneous data (Dirichlet-distributed and quantity-based label-imbalanced non-IID settings). This makes it more applicable to real-world edge devices (e.g., phones, IoT devices) than methods limited to homogeneous models.
Interesting integration of hypernetworks and distillation: The core insight—using hypernetworks for model heterogeneity while distilling synthetic data to share global knowledge—is novel. It fills a gap left by prior works: hypernetwork-based pFL methods (e.g., pFedHN) ignore data-driven signals, while federated distillation methods (e.g., FedDM) overlook model structure, and this integration addresses both heterogeneities.
Comprehensive experimental validation: Experiments cover critical dimensions to support claims: (a) Two non-IID settings (Dirichlet, quantity-based label imbalance); (b) Homogeneous/heterogeneous model architectures; (c) Generalization to held-out clients; (d) Sensitivity to IPC (images per class) of synthetic data; (e) Ablation of Contrastive Condensation Loss (L_CC) and Universum Negatives (showing explicit accuracy drops when removed); (f) Differential Privacy (DP) compatibility (only ~1% accuracy loss); (g) “Plug-and-play” gains (up to +13% accuracy) when integrating distillation into baselines; (h) Scalability to 200 clients. These results collectively confirm the method’s robustness.
Clear and accessible presentation: The paper follows a logical structure (Introduction → Related Works → Method → Experiments → Conclusion) that unfolds the framework coherently. Key components (hypernetwork backbone, synthetic data generation, local training with regularization) are explained in detail, and figures (e.g., workflow in Figure 1, feature distribution in Figure 2) effectively illustrate core mechanisms. Experiments are well-organized, making it easy to follow how results support claims.
Incremental contributions to model-heterogeneous pFL: It is among the first to integrate hypernetworks with data distillation for pFL, addressing both model and data heterogeneity from a data-driven perspective. The design of L_CC and Reg Loss (with Universum Negatives) also provides practical tools to enhance synthetic data quality and local model generalization, filling gaps in existing heterogeneous pFL methods.

**Weaknesses:**

Weak theoretical support: The paper relies solely on visualizations (e.g., Figure 2) to justify L_CC and Reg Loss, with no formal analysis of why these regularization terms balance generalization and personalization. There is no proof that L_CC-induced compact synthetic features reduce cross-client distribution shift, nor why Universum Negatives in Reg Loss enhance inter-class discriminability for personalized models. This lack of theory makes it hard to generalize the method to new datasets or architectures—undermining methodological soundness.
Inadequate privacy risk assessment: A core innovation of MH-pFedHNDD is sharing synthetic data (instead of gradients), but the paper provides no privacy analysis. There is no threat model (e.g., membership inference, attribute inference) or empirical testing to measure if synthetic data leaks more local information than gradient updates. Without this, practitioners cannot assess its suitability for privacy-sensitive domains (e.g., healthcare)—a critical oversight for federated systems.
Unquantified resource overhead: Distilling synthetic data (3000 local iterations per client) and uploading synthetic datasets introduce computational and communication costs, but these are not measured. The paper does not compare: (a) Distillation time vs. local training time; (b) Synthetic data upload size vs. gradient upload size (a key FL efficiency metric); (c) GPU/CPU memory usage for distillation on edge devices. This omits practicality checks for resource-constrained clients, weakening the method’s real-world applicability.
Lack of scenario analysis: The method shows modest gains on simple datasets (e.g., CIFAR-10, 10 classes) but significant gains on complex datasets (e.g., CIFAR-100, 100 classes; Tiny-ImageNet, 200 classes). However, the paper does not explain why this discrepancy exists—e.g., whether synthetic data adds more value for fine-grained classification (many classes) or high data heterogeneity. No explicit discussion of ideal application scenarios limits its utility for practitioners.
Weak module coupling: The three core modules—(a) hypernetwork parameter truncation (for model heterogeneity), (b) synthetic data aggregation/distribution, (c) hypernetwork co-training—operate nearly independently. Removing any one module (e.g., truncation, synthetic data) leaves a functional (though less effective) method, and there is no evidence of synergistic (“1+1>2”) benefits. For example, it does not show that hypernetwork-generated models produce better synthetic data than standalone models, nor that synthetic data improves hypernetwork parameter generation beyond local updates alone. This reduces the method’s coherence and weakens claims about its integrated design.
Scattered notation definitions: Key notations (e.g., client embedding vector v_i, model parameter count K_i, Reg Loss L_Reg) are scattered across Sections 3.3–3.5, requiring readers to cross-reference to understand their roles. This minor flaw hinders readability, even though the overall presentation is clear.
Limited transformative innovation: Hypernetworks and data distillation are existing techniques, and their combination does not introduce a paradigm shift in pFL. There are no new principles for balancing heterogeneity and collaboration—only a practical integration of existing tools. This limits the contribution’s impact compared to works that introduce novel theoretical or methodological paradigms.

**Questions:**

Regarding theoretical support for L_CC and Reg Loss: Have you conducted any theoretical analysis to formalize why these terms balance generalization and personalization? If not, could you provide empirical evidence to strengthen this link?
On privacy risks of synthetic data: Have you evaluated privacy leaks from synthetic data using standard FL attacks (e.g., membership inference, where an attacker infers if a sample was in a client’s local data)? How does the privacy risk of sharing synthetic data compare to sharing gradients—e.g.      , using metrics like membership advantage or privacy loss (ε) for DP?
About resource overhead quantification: Could you provide quantitative results on (a) the time to distill synthetic data per client (vs. local training time), (b) the size of synthetic data uploads (vs. gradient uploads for baselines like FedAvg), and (c) memory usage for distillation on edge devices (e.g., mobile GPUs)? This would clarify the method’s practicality for resource-constrained clients.
Regarding scenario suitability: Why does MH-pFedHNDD show limited gains on CIFAR-10 (10 classes) but significant gains on CIFAR-100/Tiny-ImageNet (100/200 classes)?  Is the method inherently better suited for fine-grained classification (many classes) or high data heterogeneity?
For hypernetwork parameter truncation: The paper uses truncation (θ_i = h(v_i;φ)_[1:K_i]) to support model heterogeneity, but why is this necessary?       Would a hypernetwork with architecture-specific heads (instead of truncation) achieve similar or better performance? Have you compared these two approaches to justify truncation’s value?

---

### Official Review · Reviewer_8vK2 · 2025-10-31

**Soundness:** 2
**Presentation:** 2
**Contribution:** 1
**Rating:** 2
**Confidence:** 4

**Summary:**

This paper proposes **MH-pFedHNDD** (Model-Heterogeneous personalized Federated learning framework based on HyperNetworks with Data Distillation), which aims to address both model and data heterogeneity in personalized federated learning. The key innovation is integrating data distillation into a hypernetwork-based framework. The method involves two phases: (1) a distillation phase where clients generate synthetic data using *Contrastive Condensation Loss* to create compact, discriminative synthetic datasets that are aggregated on the server; (2) a training phase where clients train personalized models generated by the hypernetwork using both local and synthetic data, guided by a *Reg Loss* with universum negatives. Experiments on CIFAR-10, CIFAR-100, and Tiny-ImageNet demonstrate improvements over existing methods.

**Strengths:**

- **Well-motivated design choices**: The two-phase framework with carefully designed loss functions (Contrastive Condensation Loss for synthetic data generation and Reg Loss with universum negatives for training) is well-motivated. The visualizations in Figure 2 effectively demonstrate how these components improve feature alignment and generalization.

- **Comprehensive experimental evaluation**: The paper conducts extensive experiments across multiple datasets (CIFAR-10, CIFAR-100, Tiny-ImageNet), various non-IID settings ($\alpha$ = 0.02, 0.05, 0.1), and both homogeneous and heterogeneous model scenarios. The generalization experiments (Section 4.2) and ablation studies provide good evidence for the method's effectiveness.

- **Consistent performance improvements on complex datasets**: MH-pFedHNDD shows consistent improvements over strong baselines, particularly on more complex datasets (CIFAR-100 and Tiny-ImageNet), demonstrating its effectiveness in challenging scenarios.

**Weaknesses:**

- **Limited novelty in technical components**: While the integration is novel, the individual technical components are largely borrowed from existing work without significant modifications:
  - The hypernetwork architecture directly follows MH-pFedHN (Zhang et al., 2025) without changes
  - $\mathcal{L}_{CC}$ is essentially SupConLoss (Khosla et al., 2020) applied to the context of synthetic data generation
  - $\mathcal{L}_{Reg}$ is directly based on UniConLoss (Han et al., 2022)
  - The data distillation method uses standard Distribution Matching (Zhao & Bilen, 2023)

- **Computational and communication overhead not analyzed**: This is a critical omission for practical federated learning:
  - No report of computational cost for the distillation phase, which requires training a separate distillation model and optimizing synthetic data
  - Communication costs not compared with baselines - while synthetic data may be compressed, the initial distillation phase and ongoing exchange add overhead
  - No wall-clock time comparisons or convergence speed analysis (only communication rounds shown)
  - Without this analysis, it's unclear whether the accuracy improvements justify the additional costs

- **Writing quality and clarity issues**:
  - Grammatical errors and inconsistent capitalization (e.g., "personalized Federated learning framework based on HyperNetworks with Data Distillation")
  - **Step 7 in Figure 1 (Distillation Phase)** is shown in the figure but never explained in the text, leaving the framework description incomplete
  - The "Difference" row in Tables 1–2 is unclear: it appears to report MH‑pFedHNDD’s gain over the second‑best method (or its gap to the best), but this should be stated explicitly in the caption or text

**Questions:**

### Q1. Distillation phase design and frequency

Several design choices regarding the distillation phase need clarification:
- **One-time vs periodic distillation**: Why is data distillation performed only once at the beginning? Intuitively, as the hypernetwork improves during training, it should generate better distillation models, leading to higher-quality synthetic data. Have you experimented with:
  - Periodic re-distillation (e.g., every K rounds)?
  - Adaptive re-distillation based on performance metrics?
  - What is the accuracy vs. distillation frequency trade-off?
- **Step 7 explanation**: What is Step 7 in the Distillation Phase shown in Figure 1? The text describes Steps 1-6 but Step 7 is clearly marked in the figure without any explanation.
- **Sensitivity to initialization**: How sensitive is the method to the quality of the initial distillation model and the architecture of the distillation model?

### Q2. Computational and communication costs

This is critical for practical federated learning deployment but is completely missing from the paper:
- **Computational overhead**:
  - How much time does the distillation phase take compared to the total training time?
  - What is the per-round training time with synthetic data vs. MH-pFedHN (without synthetic data)?
  - How many iterations/epochs are needed for distillation optimization?
- **Communication costs**:
  - What is the size of synthetic data $S_i$ uploaded by each client?
  - How does total communication cost (synthetic data + model updates) compare with MH-pFedHN and MH-pFedHNGD?
  - Is there a trade-off between IPC and communication cost?
- **Wall-clock time**: Can you provide convergence curves showing:
  - Accuracy vs. wall-clock time (not just communication rounds)
  - Time to reach certain accuracy thresholds
  - Comparison with baselines in terms of real training time

### Q3. Experimental Settings and Hyperparameter Sensitivity
- Only IPC (Images Per Class) is studied in Table 4, but critical hyperparameters $\lambda_{CC}$, $\lambda_{S}$, $\lambda_{Reg}$ are not systematically analyzed. Could author explain how these hyperparameters were chosen?
- How the number of communication rounds, local epoch, and local distillation iterations decided? Why MH-pFedHNDD performs 3000 local distillation while other methods only perform 1000 local distillation iterations?

### Q4. Synthetic data quality and characteristics

To better understand what the method learns:
- **Privacy**: If a client has very few data samples for a specific class (e.g. 50), would distilled data similar to real data? Is the proposed method safe from membership attack?
- **Visualization**: Can you provide visualizations of generated synthetic images for different clients and classes?
- **IPC trade-offs**:
  - What is the optimal IPC value across different settings?
  - Why does IPC=50 perform worse than IPC=25 (76.92% vs 77.50% at α=0.02)?
  - Is there overfitting when IPC is too large?
  - Is large IPC leakage more privacy?

---

### Note · Authors · 2025-12-30

**Comment:**

We would like to withdraw this submission. Thank you to the reviewers for their time and feedback.

**Withdrawal Confirmation:**

I have read and agree with the venue's withdrawal policy on behalf of myself and my co-authors.